# Membrane-binding mechanism of the EEA1 FYVE domain revealed by multi-scale molecular dynamics simulations

**Andreas Haahr Larsen** [ID]<sup>☯</sup>, **Lilya Tata** [ID]<sup>☯</sup>, **Laura H. John** [ID], **Mark S. P. Sansom** [ID]*

Department of Biochemistry, University of Oxford, Oxford, United Kingdom

☯ These authors contributed equally to this work.
* mark.sansom@bioch.ox.ac.uk

**Data Availability Statement:** Simulation trajectories are deposited at Zenodo: https://zenodo.org/record/5362218#.YTDSpS0Rpqs (DOI: 0.5281/zenodo.5362218).

## Abstract

Early Endosomal Antigen 1 (EEA1) is a key protein in endosomal trafficking and is implicated in both autoimmune and neurological diseases. The C-terminal FYVE domain of EEA1 binds endosomal membranes, which contain phosphatidylinositol-3-phosphate (PI(3)P). Although it is known that FYVE binds PI(3)P specifically, it has not previously been described of how FYVE attaches and binds to endosomal membranes. In this study, we employed both coarse-grained (CG) and atomistic (AT) molecular dynamics (MD) simulations to determine how FYVE binds to PI(3)P-containing membranes. CG-MD showed that the dominant membrane binding mode resembles the crystal structure of EEA1 FYVE domain in complex with inositol-1,3-diphospate (PDB ID 1JOC). FYVE, which is a homodimer, binds the membrane via a hinge mechanism, where the C-terminus of one monomer first attaches to the membrane, followed by the C-terminus of the other monomer. The estimated total binding energy is ~70 kJ/mol, of which 50–60 kJ/mol stems from specific PI(3)P-interactions. By AT-MD, we could partition the binding mode into two types: (i) adhesion by electrostatic FYVE-PI(3)P interaction, and (ii) insertion of amphipathic loops. The AT simulations also demonstrated flexibility within the FYVE homodimer between the C-terminal heads and coiled-coil stem. This leads to a dynamic model whereby the 200 nm long coiled coil attached to the FYVE domain dimer can amplify local hinge-bending motions such that the Rab5-binding domain at the other end of the coiled coil can explore an area of 0.1 μm<sup>2</sup> in the search for a second endosome with which to interact.

## Author summary

Peripheral proteins bind to the surfaces of specific membranes within eukaryotic cells and thereby coordinate dynamic interactions between sub-cellular compartments. The FYVE domain of EEA1 recognizes the membranes of endosomes by binding to the anionic headgroup of a specific lipid, phosphatidylinositol-3-phosphate (PI(3)P), which is present in those membranes. We use molecular dynamics simulations at two different resolutions (coarse-grained and all atom) to define the mechanism whereby a dimeric FYVE domain binds to model membranes containing PI(3)P. Hinge-bending flexibility between the

**Funding:** AHL was funded by the Carlsberg foundation (CF19-0288), https://www.carlsbergfondet.dk/en. LHJ was funded jointly by the BBSRC Interdisciplinary Bioscience DTP(BB/M011224/1), https://bbsrc.ukri.org/. MSPS was funded by Wellcome (208361/Z/17/Z), https://wellcome.org/; BBSRC (BB/R00126X/1), https://bbsrc.ukri.org/; and EPSRC (HECBioSim, EP/R029407/1), https://epsrc.ukri.org/. The funders had no role in study design, data collection and analysis, decision to publish, or preparation of the manuscript.

**Competing interests:** The authors have declared that no competing interests exist.

bound FYVE domain and the coiled coil stalk to which it is attached is observed. In the intact EEA1 molecule the coiled coil is 200 nm long. This 'amplifies' the flexibility observed at the FYVE hinge end such that the other end of the EAA1 protein can sweep out an area of about 0.1 μm$^2$ in its search for a second endosome with which to interact. In this way EAA1 acts as a long stalk with two 'sticky' ends to help draw together two specific membrane surfaces.

## Introduction

Early Endosomal Antigen 1 (EEA1) acts as a connecter between the early endosomal membrane and Rab GTPase effector proteins [1,2]. Thereby, it plays a key role in endosomal trafficking. Autoantibodies to EEA1 have been implicated in several autoimmune diseases, such as subacute cutaneous systemic lupus erythematosus, polyarthritis and rheumatoid arthritis [3,4]. EEA1 is also implicated in neurological diseases [5] and therefore provides a potential drug target.

EEA1 is homodimeric, composed of a FYVE (Fab1, YOTB, Vac1, EEA1) domain located at the C-terminus of a ~200 nm long coiled-coil. At the N-terminus of the coiled coil, there is a Rab5-binding domain [6–8] (Fig 1). The FYVE domain interacts with the cytosolic face of early endosomal membranes, which contain phosphatidylinositol-3-phosphate (PI(3)P) [9]. Dimeric FYVE domain binds PI(3)P [9–11]. The binding site was identified by X-ray

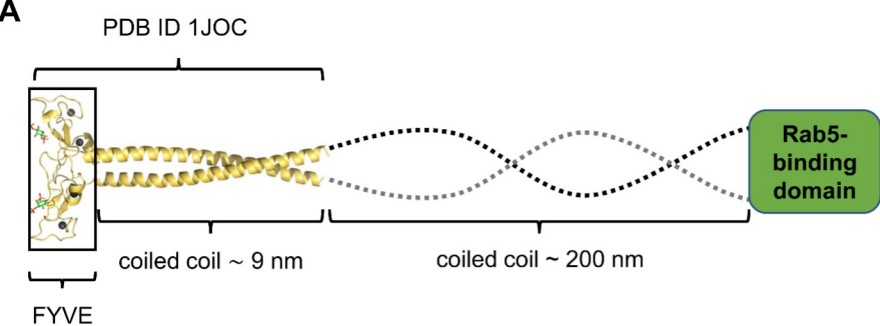

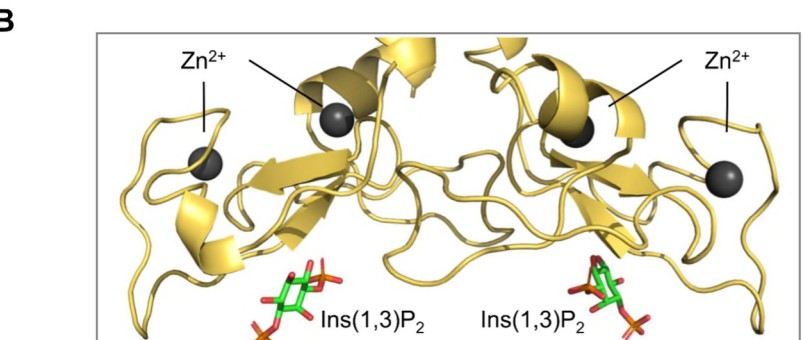

**Fig 1. Structure of EEA1 and the FYVE domain.** (A) Schematic of the full-length structure of dimeric EEA1, with the crystal structure of the FYVE domain dimer plus start of the coiled coil (PDB ID 1JOC) in yellow, followed by a ~200 nm coiled coil (dotted line) and a Rab5-binding domain. (B) The FYVE domains of the dimer, with four zinc ions (dark spheres) bound by 4-cysteine zinc fingers. Two bound inositol (1,3)-bisphosphate molecules (red/orange/green) are shown.

crystallography of Ins(1,3)P$_2$ bound to dimeric FYVE (PDB: 1JOC) [7] and by NMR of mono-meric FYVE in a complex with Ins(1,3)P$_2$ (PDB: 1HYI) [6,12,13].

The interaction of a FYVE domain with a membrane containing PI(3)P molecules has pre-viously been investigated with molecular dynamics (MD) simulations [14], demonstrating the short timescale (30 ns) stability of monomeric domain docked onto a membrane in an orienta-tion consistent with experimental results [6,7,12]. There have been a number of developments in MD simulations which enable improved sampling of protein/membrane interactions, both at the atomistic [15–17] and coarse-grained (CG) level [18–22]. We have simulated binding of a larger dimeric construct of FYVE in 1.5 μs coarse-grain simulations followed by a 500 ns atomistic simulation to refine the bound pose. In these simulations the dimeric FYVE domain was positioned in different orientations at least 10 nm above the membrane, thus avoiding bias in sampling of possible binding modes of the protein. Analysis of these simulations provides a detailed and dynamic model of how dimeric FYVE binds to target membranes.

## Results and discussion

### CG-MD simulations of FYVE binding to membranes

Coarse-grained MD simulations were performed to investigate the binding of the dimeric EEA1 FYVE domain to a simple model of an endosomal membrane composed of PC (95%) plus 5% PI(3)P which is known to bind FYVE. A truncated version of the crystal structure of EEA1 FYVE (PDB 1JOC) was used, with a shorter coiled-coil stalk (see Methods for details). Fifteen repeats of the simulation, each of duration 1.5 μs, were performed with different ran-domly selected starting orientations of the FYVE with respect to the membrane, and with a minimum distance of 10 nm between protein and membrane (Figs 2A and S1). The minimum distance between the protein and the bilayer was monitored as a function of time for each sim-ulation (Fig 2B). In 8/15 repeats, the FYVE domain bound to the membrane within the 1.5 μs of the simulations. In the remaining 7/15 repeats, the FYVE domain did not approach close enough to the bilayer to experience an electrostatic interaction, i.e. the minimum distance remained larger than the cutoff distance (see Methods), and so no binding event occurred. For

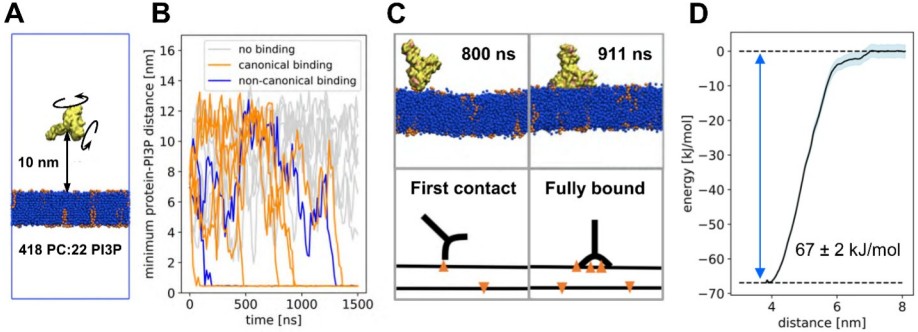

**Fig 2. Coarse-grained simulations.** (A) Set-up of the CG simulation, with the truncated FYVE dimer (yellow) placed at a minimum distance of 10 nm from the lipid bilayer (blue; orange PI(3)P molecules) and rotated at random. (B) Minimum distance for each of the 15 repeats, showing repeats without binding (grey, 7/15), with canonical binding (similar to the binding mode of the crystal structure, orange, 6/15) and with non-canonical binding (blue, 2/15). (C) Two snapshots from one of the repeats with canonical binding, after 800 and 911 ns. The snapshots show the hinge-mechanism by which FYVE binds to the membrane. Sketches below for clarity. (D) Potential of mean force (PMF) calculations for the binding of FYVE to a lipid membrane containing PC and PI(3)P, in the canonical binding mode, showing mean (dark blue line), standard deviation (shaded area) as well as minimum and maximum values (dotted lines). The PMF well depth is 67 ± 2 kJ/mol.

6 of the binding events, the FYVE domain bound in a 'canonical' manner, with the C-termini of each chain bound to the membrane (Fig 2B), whereas two resulted in non-canonical binding (Figs 2B and S2). Intriguingly, we observed a two-stage mechanism of binding, for the canonical binding events, where FYVE first attached to the membrane via the C-terminus of one chain, then made a rotation around the point-of-contact such that the second C-terminus bound as well (Fig 2C). Once this canonical binding mode was established, the FYVE dimer remained in this pose for the duration of the simulation. In control simulations using a PC bilayer with no PI(3)P present, none of the simulations resulted in a canonical interaction. Of the 15 PC-only repeats, 8 showed non-canonical binding, of which 3 subsequently dissociated from the membrane (S3 Fig).

Returning to the FYVE dimers bound to the PI(3)P membrane in a canonical pose, these often revealed between 5 to 10 PI(3)P molecules interacting with the FYVE dimer in this binding pose, in contrast to two $Ins(1,3)P_2$ molecules bound in the dimeric crystal structure (PDB: 1JOC; one bound per monomer). The NMR structure of the FYVE monomer also had just one $diC_4$-PI(3)P molecule bound (PDB: 1HYI) [13]. We suggest that multiple PI(3)P lipids are likely to interact with FYVE *in vivo*, but only one binding site/monomer has sufficiently high affinity to retain bound PI(3)P in the absence of a membrane. This is in line with previous studies of other PIP-binding domains, e.g. PH [23] and C2 [24], which interact with multiple PIPs and other anionic lipids.

## Strength of FYVE/PI(3)P interactions

To investigate the strength of the predicted FYVE/PI(3)P interactions, we undertook both potential of mean force (PMF) and free energy perturbation (FEP) calculations to estimate the free energy of PI(3)P interactions with the protein as estimated by the CG forcefield [25,26]. For the energy calculations, a representative frame of the bound structure in a canonical pose, with six PI(3)P molecules interacting with the dimeric FYVE domain was used. A PMF, using the bilayer to protein centre of mass distance as a reaction coordinate, was estimated by umbrella sampling (Fig 2D). The depth of the energy well, corresponding to the membrane bound FYVE dimer interacting with six PI(3)P molecules, was -67 ± 2 kJ/mol. The free energy of binding must, however, be found by integration over the PMF (see Methods). This leads to a slightly larger free energy of binding of -74 kJ/mol. The free energy associated with the headgroup of each bound PI(3)P molecule was calculated via FEP calculations [25], whereby each of the six bound PI(3)P headgroups were changed into PC headgroups (S4 Fig). The free energy of this perturbation varied between 6 and 12 kJ/mol for each PI(3)P molecule converted to PC, summing to a total perturbation energy of +55 ± 2 kJ/mol (S4A Fig). Thus, the InsP headgroups comprise about 75% of the total membrane binding free energy as estimated by the PMF. Furthermore, the FEP calculations suggest a twofold range in interaction free energies for the different PI(3)P sites.

The dissociation constant ($K_d$) of a EEA1 FYVE monomer with a PI(3)P-containing membrane has been estimated experimentally to ~50 nM [27] at room temperature, corresponding to a binding free energy of -42 kJ/mol. If we make the simple assumption that the binding energies are additive, when going from monomer to dimer, this corresponds to -84 kJ/mol for the dimer, i.e. slightly greater than the PMF from the simulations. A later study has shown that $K_d$ is pH-dependent: ~50 nM (42 kJ/mol) at pH 6.0; ~500 nM (36 kJ/mol) at pH 7.4; and ~1200 nM at pH 8.0 (34 kJ/mol) [28]. Again, these numbers are for monomeric FYVE. Given the latter, it would be attractive to calculate the free energy of dimerization of FYVE, but this is made rather challenging by the involvement of the coiled coil stalk in dimer stabilization. The protonation states in our simulations are designed to match the conditions at neutral pH. We

note that the free energy estimated from the PMF, -74 kJ/mol is consistent with two times the value reported for the monomer at pH 7.4 (-72 kJ/mol) [28].

We further analysed the binding sites using the program PyLipID [29], which can identify protein-lipid binding sites based on residence time in CG-MD simulations. The three main binding sites for PI(3)P on the dimeric FYVE domain were found to be (S4 Fig): (i) residues 1348–1352 (KWAED), where residues 1349–1352 (WAED) form a conserved WxxD motif, and where a W1349A mutation has been shown to substantially reduce PI(3)P binding [27]; (ii) a second site made up primarily of residues 1397–1400 (KPVR), where R1400 is part of a conserved RVC motif of FYVE, mutation of which (R1400A) leads to 6-fold decrease in PI(3)P binding affinity [27]; and (iii) a third site which includes residues 1368–1373 (TVRRH) at the core of a conserved motif spanning residues 1371 to 1376 (RRHHCR), and mutations of which lead to impaired PI(3)P affinity [27]. The first and second binding sites are both placed centrally (S4 Fig), just below the stem of the FYVE domain, similar to the binding sites from the crystal structure complexed with Ins(1,3)P2 (PDB ID 1JOC) [7] and the NMR structure, complexed with diC4-PI(3)P (PDB ID 1HYI) [13].

## Atomistic MD simulations

Using atomistic MD simulations, we analysed the mode of interaction of the FYVE dimer with the membrane in more detail. Starting from a representative pose of the membrane-bound protein from one of the CG simulations (Fig 3), we simulated three repeats, each 0.5 μs in length. The AT-MD simulation showed that the two "heads" (i.e. C terminal domains) of the FYVE dimer did not remain equally close to the membrane throughout the duration of the simulations but was slightly tilted with respect to the bilayer (Fig 3). The COM distance for each head was monitored. In the first two repeats, head 1 (C terminus of the first monomer) was systematically further away from the membrane than head 2 (Figs 4A and S5). In repeat 3, the order changed during the simulation, as anticipated from the symmetry of the dimeric protein. However, the tilted binding orientation was conserved, with one head closer to the membrane than the other. The number of H-bonds to PI(3)P lipids was seemingly independent on the COM distance (Figs 4B and S5) across the repeats. Hence, we suggest two modes of interaction for the FYVE heads: (i) mode 1, where the head is relatively far from the membrane and H-bond formation is the major component of binding and (ii) mode 2, where the head is closer to the membrane, with a loop inserted into the bilayer (Fig 3).

In the coarse-grain simulations, secondary and tertiary structure was restrained with an elastic network (see Methods). This was not the case in the AT simulations, and so possible large scale conformational changes could thus be observed. From analysis of RMSDs (S6 Fig) we noted that the heads and likewise the coiled coil (stalk) behaved individually as semi-rigid bodies, as reflected in Cα RMSDs of 0.30 ± 0.03 nm for the heads and 0.47 ± 0.05 nm for the stalk (S6 Fig). However, there was substantial hinge-bending flexibility between heads and stalk.

The hinge-bending motion could be observed via visual inspection (Fig 3). By calculating the angle between the stalk axis and the plane of the bilayer (Figs 4C and S5) we can see the local consequence of these fluctuations, with tilt angles up to ~50˚. This could be even greater if one allows for bending of the coiled coil, and indeed recent evidence [30] suggests that binding of Rab5:GTP may increase the flexibility of the EEA1 coiled coil. We note that the angle varies among the repeats and the maximum angle of around 50˚ is merely an approximate value. As the coiled coil in the full-length EEA1 protein is more than 200 nm long, the flexibility of the link of early endosome PI(3)P-bound FYVE dimer to the coiled coil would provide a search area of at least 0.09 μm$^2$ at the other end of the EAA1 molecule for a Rab5 molecule attached to a second early endosome (Fig 5). This emphasizes the importance of considering

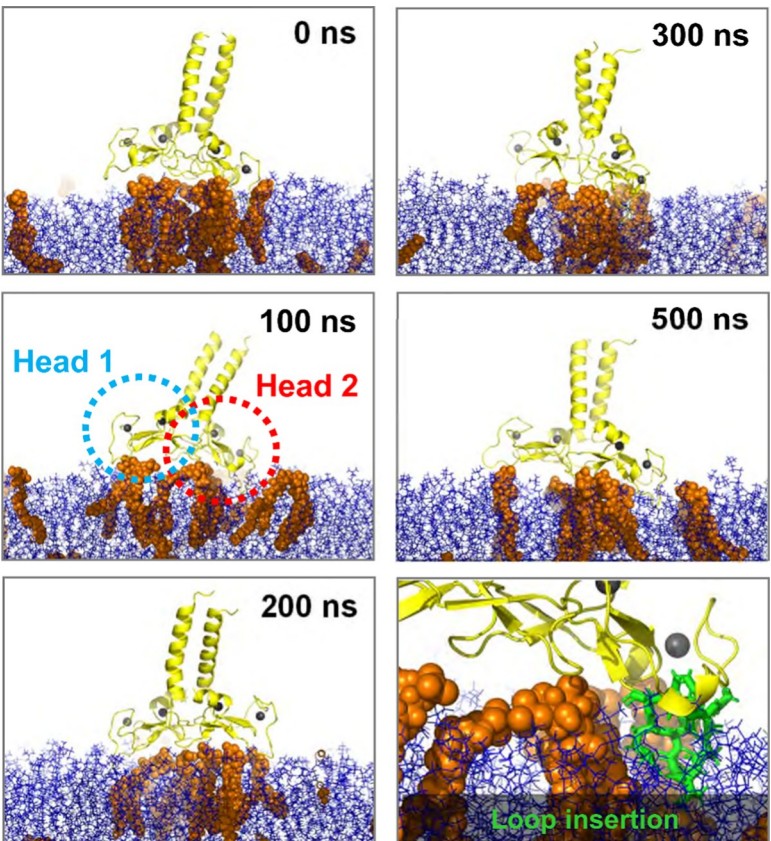

**Fig 3. Atomistic simulations.** Snapshots from a representative simulation at 0, 50, 100, 200, 300 and 500 ns. Heads 1 and 2 of the FYVE domain are highlighted (dotted circles) for the 100 ns snapshot. The inserted loop as seen at 100 ns shown in in green stick representation in the final panel.

molecular flexibility when relating static domain structures to dynamic tethering within a cell. We conjecture that mutations in this hinge region could modulate the flexibility, and thus alter the ability of early endosome PI(3)P-bound EEA1 to capture adjacent endosomes.

## Conclusions

We have investigated the membrane-binding model for FYVE proposed by e.g. Dumas et al. [7]. Using MD simulations, we have verified that the dominant binding mode is with the C-terminus of each chain of the homodimeric FYVE domain facing the membrane. We have described different modes of binding for the FYVE domain, either driven by electrostatic interactions with PI(3)P headgroups, or driven by insertion of loops in the hydrophobic bilayer. We have shown that FYVE binds multiple PI(3)Ps, albeit more weakly to some of them. Moreover, we have shown that there is hinge-bending flexibility in the linker between the PI(3)P-bound FYVE domain heads and the coiled-coil stalk, thus facilitating search and capture of adjacent early endosomal membranes. A future challenge will be to build a (CG) model of a complete EAA1 molecule linking two vesicles to more fully explore the role of hinge-bending and coiled-coil flexibility in vesicle tethering. This would be a large simulation but not impossible (see e.g. [31,32]).

From a more methodological perspective it is important to understand to what extent the use of CG simulations influences e.g. the probability of observing different binding modes of the FYVE domain dimer to PIP-containing membranes. Possible limitations of the Martini 2

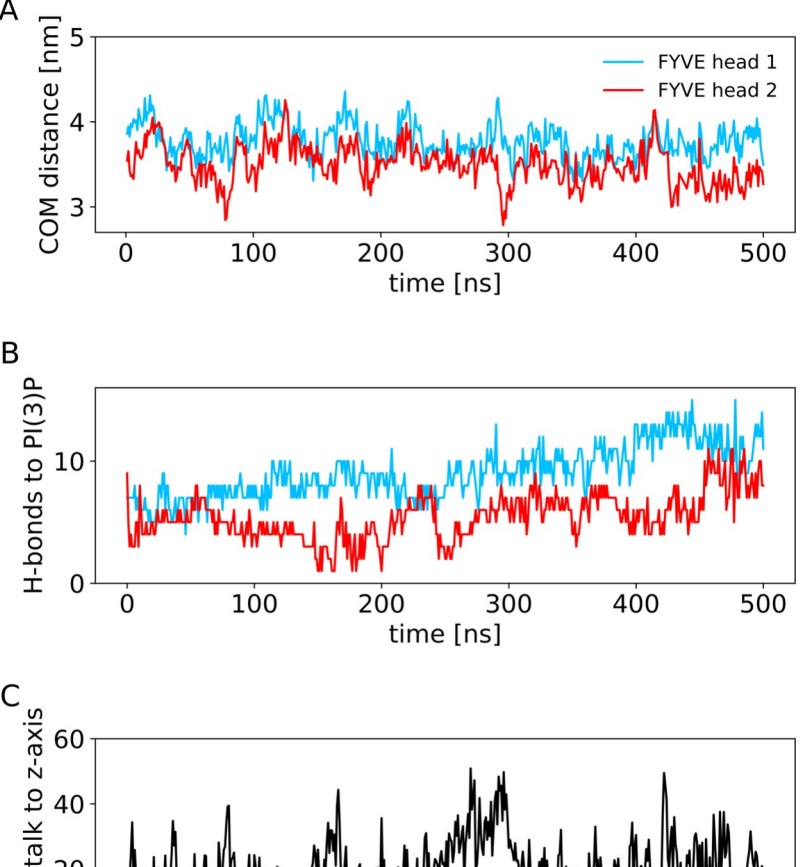

**Fig 4. Atomistic simulations: distances, H-bonds and stalk tilt angle.** (A) Centre-of-mass distance between the membrane and each of the FYVE heads as function of time. (B) Number of H-bonds with PI(3)P headgroups for the FYVE heads. (C) Angle between the stalk axis and the bilayer normal as a function of time.

model used in the current CG simulations have been addressed with the release of Martini 3 [33]. However, the new version of the CG forcefield will require careful evaluation using well documented test systems (e.g. PH domains [34]) of the Martini 3 parameters for protein/PIP interactions [35].

Overall, our simulations provide a dynamic model of how FYVE binds to target PIP-containing membranes to facilitate the tethering of early endosomes. This interaction is brought about by a dimeric recognition domain connected to a locally rigid stalk via a flexible hinge. The stalk is part of a ~200 nm long coiled coil which amplifies local hinge-bending motions such that the Rab5-binding C2H2 finger at the opposite end of the coiled coil can explore an area of ~0.1 $\mu m^2$ in the search for a second endosome with which to interact.

## Methods

### Coarse-grained simulations

The crystal structure of the EEA1 FYVE domain in complex with 1,3-diphosphate (PDB ID 1JOC) was used to generate the initial structure for the simulations (Fig 1). The wildtype EEA1

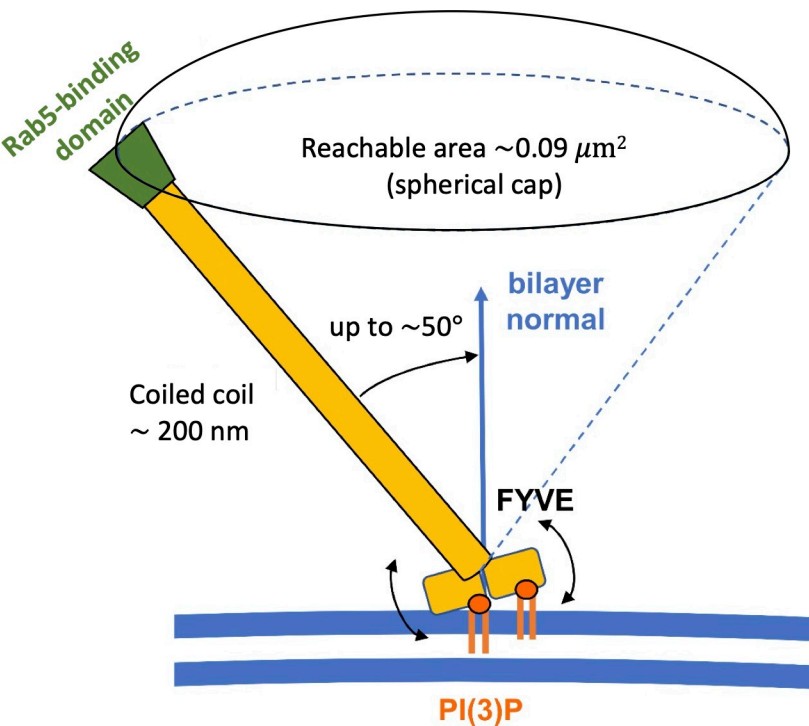

**Fig 5. Vesicle tethering and title of EAA1 coiled coil stalk relative to the bilayer normal.** The FYVE dimer is shown interacting with two PI(3)P molecules (orange) in a cell membrane (blue lines). Based on the analysis of our simulations this allows the coiled coil to tilt by up to 50˚ relative to the bilayer normal. Thus, the Rab5-binding domain (green) is able to search an area of ~0.09 μm² for a contact to the target vesicle.

has a ~200 nm long coiled-coil stalk, running from residues 74 to 1348, followed by a FYVE domain (residues 1352–1411). The crystal structure of the dimer (1JOC) has a truncated coiled-coil stalk formed by residues 1289–1348. Initial simulations of this construct resulted in biologically unrealistic interactions of the cationic N-terminal region (residues 1289–1322) of this stalk with the bilayer. We therefore further shortened the stalk in the simulations so that it started at residue 1323, which resulted in biologically realistic interactions with the bilayer via the FYVE domain rather than the stalk. The truncated construct was coarse-grained using the Martinize script [36] to a construct compatible with the Martini2.1 force field [37]. Zinc ions and bound 1,3-diphosphate were removed during the coarse-graining of the initial structure. An elastic network [38] was applied to conserve secondary and tertiary structure, which maintained the CG protein fold in the absence of explicit zinc ions. A simulation box with the coarse-grained protein, water, ions and a lipid membrane was constructed using the insane script [39]. The initial box size was 16x16x32 nm³ and had periodic boundary conditions. 10% anti-freeze water were added to avoid frozen water [37], and 0.15 M NaCl ions was included. The protein was placed with a minimum distance of 10 nm from the membrane to avoid any initial binding bias (Figs 2A and S1). The membrane was symmetric and had a lipid composition of 95% 1-palmitoyl-2-oloyl-sn-glyecero-3-phosphocholine (POPC) [39] and 5% 1-palmitoyl-2-oloyl-sn-glyecero-3-phosphatidylinositol-3-phosphate (POP(3)P) [20], giving a lipid composition of 418:22 (POPC:POP(3)P) in each leaflet.

Simulations were performed in GROMACS 2018 [40]. The protein was first rotated randomly, then minimised with the steepest decent method, equilibrated for 300 ps and finally simulated for 1500 ns. Both equilibration and production were in the NPT ensemble with a

semi-isotropic Parinello-Rahman barostat [41] with 12 ps time constant. Lateral and normal compressibility of $3 \cdot 10^{-4}$ bar$^{-1}$ was used to keep pressure at 1.0 bar, and a v-rescale thermostat with a 1 ps time constant kept the temperature at 323 K. The integration timestep was 30 fs. Long range electrostatics were calculated using particle mash Ewald (PME) [42] with a cut-off of 1.1 nm. Non bonded van der Waals bonds likewise had a 1.1 nm cut-off. The initial position of the protein had a minimum protein-lipid distance of 10 nm, i.e. multiple times the cutoff-distances. 15 repeats were made, each with individual rotation, minimisation and equilibration. The minimum protein-membrane distance was monitored using built-in GROMACS tool *gmx mindist*. Lipid contacts were analysed using PyLipID [29]. Contacts were measured from their residence time [43].

## Potential of mean force estimation

A frame from the coarse-grained simulations was selected for calculations of free energies, corresponding the dominant membrane binding mode and with 6 bound PIP(3)P molecules. For the PMF calculations, the protein was pulled away from the membrane using steered MD and a pulling force of 1000 kJ/mol/nm, along a reaction coordinate perpendicular to the membrane plane, employing the GROMACS pull code. The reaction coordinate was the centre-of-mass distance between the lipid bilayer and the protein. Position restraints were applied to PI(3)P heads to prevent these from being pulled out of the membrane. Snapshot were retrieved every 0.5 nm along the reaction coordinate. These were used for umbrella sampling, consisting of 1000 ns simulations with an applied position restraint of 1000 kJ/mol/nm on the protein. From the umbrella sampling, a 1D energy profile was calculated using the built-in weighted histogram method *gmx wham* [44], employing 200 rounds of bootstrapping.

The dissociation constant can be determined from the PMF by integration [45–47]:

$$K_d = 4\pi C^\circ \int_{r_b}^{r_c} e^{-PMF(r)/(RT)} r^2 dr,$$

where $C^o$ is the standard concentration (eliminated in the next equation), $r_b$ is the binding distance (at the minimum of the PMF), and $r_c$ is the distance at which FYVE does not "feel" the membrane anymore, i.e., the bulk, where the PMF has converged. Thus, $r_b$ defines the "range" of the binding interaction and thus affects the free energy of binding. The binding free energy is given by:

$$\Delta G_{bind} = -RT\log(K/C^\circ).$$

We used $T$ = 300 K, $r_b$ = 3.9 nm, and $r_c$ = 7.0 nm to calculate the reported binding free energy.

## Free energy perturbation

In six independent simulations, one bound PI(3)P headgroup was alchemically transformed into a PC headgroup, by changing the CG bead properties. First the Coulombic interactions were perturbed in 20 steps of 100 ps, and in 20 subsequent steps, the Lennard Jones properties were perturbed. One simulation was also run with free PI(3)P being transformed into PC, and free energy was calculated as the difference between the perturbation energy of bound and free PI(3)P [25], using the alchemical analysis package [48].

## Atomistic simulations

A frame from the coarse-grained simulations corresponding the dominant membrane binding mode and with 6 bound PIP(3)P molecules (as used for energetic calculations–see above) was

converted using CG2AT [49] to atomistic resolution and prepared for simulations in the CHARMM36m force field with TIP3P water. $Zn^{2+}$ ions were inserted in the zinc-finger sites in accordance with the crystal structure (PDB: 1JOC). The four cysteines in each zinc fingers were converted to deprotonated cysteine (topology CYM). Zinc ions were restraint with 1000 kJ/mol/nm$^2$ bonds to the neighbouring cysteines using the pull code in GROMACS. The system was neutralised by converting four $Na^+$ ions to $Cl^-$. The system was equilibrated for 10 ns in the NVT ensemble and for 20 ns in the NPT ensemble, then simulated for 500 ns. Both equilibration and production run were kept at 300 K using a v-rescale thermostat with a 0.1 ps time constant. NPT equilibration and production run were kept at 1.0 bar using a semi-isotropic Parrinello-Rahmen barostat [41] with a 5 ps time constant and $4.5 \cdot 10^{-4}$ bar$^{-1}$ isothermal compressibility. LINCS constraints were applied on the H-bonds, so the simulation could be run with 2 fs time steps with a leap-frog integrator. Electrostatics were included using particle mesh Ewald (PME) [42] for long range electrostatics, and a cut-off of 1.2 nm. The van der Waals cut-off for non-bonded interactions was also 1.2 nm. H-bonds were analysed using the *gmx hbonds* tool in GROMACS. COM distances were calculated using *gmx distance*, with the distance defined between the lipids and the protein as separate groups.

Visualisation of CG simulations used VMD [50], and of atomistic simulations used PyMOL (The PyMOL Molecular Graphics System, Version 1.2, Schrödinger, LLC). Scripts and input files for the simulations are available via github.com/andreashlarsen/Larsen-Tata2021-FYVE (DOI: 10.5281/zenodo.5048289). Simulation trajectories are deposited at Zenodo: https://zenodo.org/record/5362218#.YTDSpS0Rpqs (DOI: 10.5281/zenodo.5362218).

## Supporting information

**S1 Fig. Initial CG-MD snapshots.** Snapshots from the 15 repeats of the CG-MD, taken after equilibration. FYVE (yellow) is in a random orientation above the membrane composed of POPC (blue) and POPI(3)P (orange).
(TIF)

**S2 Fig. Non-canonical binding modes.** Snapshots of the non-canonical binding modes seen in repeat 11 and repeat 14. FYVE (yellow) bound to the lipid membrane composed of POPC (blue) and POPI(3)P (orange).
(TIF)

**S3 Fig. Coarse-grained simulations of the association of the truncated FYVE dimer with a bilayer containing only PC lipids.** The minimum distance between the protein and bilayer is shown as a function of time for each of the 15 repeat simulations, showing no binding (blue, 7/15), non-canonical binding followed by dissociation (red 3/15), or non-canonical binding (yellow, 5/15).
(TIF)

**S4 Fig. PI(3)P contacts.** (A) Table of free energies for conversion of PI(3)P to PC at 6 selected sites (shown in (B) where the FYVE dimer, yellow, is viewed from the membrane facing side) where PI(3)P was seen bound in CG simulations. The table lists the corresponding FEP estimates for PI(3)P conversion to PC at each site. (C) Crystal structure (PDB ID 1JOC) of the FYVE dimer with two bound inositol (1,3)-biphosphates (red/orange/grey) and zinc ions (blue/purple spheres). (D) Binding sites on the FYVE dimer structure from CG simulations with the two most prominent binding sites on each monomer shown as coloured spheres. (E) Sequence of chains A and B from dimeric FYVE, highlighting the lipid contact sites using the same residue colours as in panel D, with conserved sequence motifs underlined.
(TIF)

**S5 Fig. 500-ns AT-MD, repeats 2 and 3.** (A-C) repeat 2, (D-F) repeat 3. (A, D), centre of mass distance between each FYVE head and the membrane. (B, E) Number of H-bonds between each FYVE head and PI(3)P headgroups. (C, F) Angle between the coiled coil stalk axis and the membrane normal.
(TIF)

**S6 Fig. RMSD analysis of rigid body motions in AT-MD simulations.** Cα RMSDs from the atomistic simulations are shown for the FYVE dimer heads aligned to the initial structure of the heads (black), for the coiled-coil stalk aligned to the stalk (purple), and for the heads with the dimer structures aligned to the stalk (gold). This demonstrates the rigid body motions between stalk and heads observed during the simulation.
(TIF)

## Acknowledgments

The authors would like to thank Robin Corey for input on running free energy calculations, Owen Vickery for extending the CG2AT script, and Sarah-Beth Amos for providing useful scripts for the analysis.

## Author Contributions

**Conceptualization:** Andreas Haahr Larsen, Lilya Tata, Mark S. P. Sansom.

**Funding acquisition:** Andreas Haahr Larsen, Mark S. P. Sansom.

**Investigation:** Lilya Tata, Laura H. John.

**Methodology:** Andreas Haahr Larsen, Lilya Tata, Mark S. P. Sansom.

**Project administration:** Mark S. P. Sansom.

**Supervision:** Andreas Haahr Larsen, Laura H. John, Mark S. P. Sansom.

**Visualization:** Lilya Tata.

**Writing – original draft:** Andreas Haahr Larsen.

**Writing – review & editing:** Laura H. John, Mark S. P. Sansom.

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
