## [Decision Letter · Decision Letter 0]

20 Mar 2021

Dear Dr. Sansom,

Thank you very much for submitting your manuscript "Membrane-binding mechanism of the EEA1 FYVE domain revealed by multi-scale molecular dynamics simulations" for consideration at PLOS Computational Biology.

As with all papers reviewed by the journal, your manuscript was reviewed by members of the editorial board and by several independent reviewers. In light of the reviews (below this email), we would like to invite the resubmission of a significantly-revised version that takes into account the reviewers' comments. Reviewer #1 in particular has several comments where clarification would be useful.

For the revised submission, it is also important that all data and input files necessary to reproduce the work is available in a public repository (such as Zenodo) according to the PLoS data sharing policies, and that the article contains a link to this repository.

We cannot make any decision about publication until we have seen the revised manuscript and your response to the reviewers' comments. Your revised manuscript is also likely to be sent to reviewers for further evaluation.

Sincerely,

Erik Lindahl

Guest Editor

PLOS Computational Biology

Nir Ben-Tal

Deputy Editor

PLOS Computational Biology

Reviewer's Responses to Questions

**Comments to the Authors:**

Reviewer #1: The manuscript by Sansom and co-workers describes atomistic and coarse-grained (CG) simulations of the C-terminal FYVE domain of EEA1 in the presence of membranes mimicking an endosome. The subject is of high biological interest, as the protein is involved in several diseases. The methodology is well-established and appropriate for the problem at hand. The manuscript is very clear in all parts. The general conclusions are supported by the data. I very much enjoyed reading the manuscript, however I would like to indicate a few points that should be improved:

1. On calculating the protein-lipid binding affinity from a PMF: the correct way to do it was established by Shoup and Szabo (Biophys. J. 40, 33–39 (1982)), and has been reported by several others (see, for example, eq. 44 in DOI:10.1002/jcc.21776). Using the correct equations, the results will probably be similar - most likely this will not change dramatically the results.

2. Page 6: "As the coiled coil in the full-length EEA1 protein is more than 200 nm long, and will be locally rigid, a tilt of up to ~50° relative to the bilayer normal would result in a later displacement of the Rab5 binding domain at the opposite end of nearly 2000 nm". I cannot follow the math. It seems to me that, if the stalk is about 200 nm in length, and Rab5 is attached to the opposite end of the FYVE heads, tilting the stalk by whatever amount would move Rab5 by 200 nm at maximum (provided that the stalk does not go through the membrane, does not unfold, and behaves as a rigid body). I also do not follow the math for the reachable area, that seems to be pi*(200 nm * (sin theta))^2 if theta is small (i.e., one can assume that the searchable area is a circle). If theta is large, the searchable area should be represented as a spherical cap [ area = 2*pi*r^2(1 - cos theta) ]. In any case, with theta = 50 degrees, the result I get for the searchable area is much less than 0.12 um^2.

Other points:

- On protein-lipid binding modes explored in AA simulations: it is apparent that the sampling achieved in the AA simulations is insufficient to assess the binding modes in a quantitative way. That is expected, considering the kinetics of protein binding/unbinding. I wonder if differences in binding modes were observed also in CG simulations. In that case, extension of the CG simulations would allow a more quantitative assessment of the probability of the different binding modes. If instead the CG model is not accurate enough in this respect, it would be interesting to determine which features/ingredients are missing in the CG model to enable mimicking the AA binding modes (electrostatics or hydrogen bonding too primitive? Shape of the binding pockets too different/smooth compared to AA? Other reasons?). In general, it would be interesting to make a more quantitative comparison between AA and CG binding modes.

- On the hinge-bending motion observed in AA simulations: it is very unlikely that AA simulations converged or explored all thermodynamically accessible states in such short amount of simulation time (although, of course, it is recognised that these are computationally very expensive simulations). At the same time, it is also unlikely that the CG model will reproduce such motion realistically - because it is difficult to calibrate the elastic network. Perhaps a word of caution on the AA results should warn the reader that the 50 degree estimate is just an educated guess.

Reviewer #2: Larsen & Tata et al. present a computational study on how the FYVE domain of EEA1 binds to PI3P-containing membranes such as those in endosomes. The computational work is solid (I only have one comment regarding the atomistic simulation), and the paper is clearly organized.

Main comments:

- Was there only one atomistic simulation? I think the atomistic simulation should be repeated at least two more times (or more, depending on the results of the two additional simulation runs), as it is hard to draw conclusions from a single simulation.

- It would be great to see the 15 starting orientations for the coarse-grained simulations in a supplemental figure.

- In the Methods: "This crystal structure is positively charged at its N-terminal region (i.e. the start of the coiled coil stalk, residues 1289-1322 of EEA1), which resulted in artefactual binding of the N-terminus to the anionic lipid bilayer membrane. Therefore, we used a truncated version of the protein with a shorter coiled-coil stalk consisting of residues 1323-1411."---This is a bit confusing; please clarify exactly how the simulated system differs from the crystal structure and from wildtype EEA1.

- I think it would be good to briefly comment on the more general biological significance of the findings in the conclusions and/or abstract.

Minor comments:

- In the abstract, I would change "... total binding energy is ..." to "estimated total binding energy" or similar

Reviewer #3: The paper describes an interesting attempt to simulate the binding of the EEA1 FYVE domain binds to endosomal membranes. The simulations are done by multiscale approach and seems to give reasonable results.

I only have a few comments.

1)I am not sure about the assessment of the dimerization effect. This should involve calculations of the dimerization free energy. This issue should be addressed.

2) Other approaches for CG and MD simulations of insertion of membrane proteins into membranes should have been mentioned and discussed.

Including , Roux , Warshel (CG) , Ben-tal, Lindahl and others

Reviewer #4: This manuscript describes atomistic (AT) and coarse-grained (CG) simulations aimed at understanding how the protein EEA1, specifically its C-terminal FYVE domain, binds to PI-containing membranes. From the CG simulations, they find that binding occurs via a hinge mechanism with a total free-energy gain of 67 kJ/mol, in reasonable agreement with experiments. AT simulations of the bound-state resolves binding into two types as well as demonstrates significant flexibility of the stem domain.

I found this manuscript very clear and well reasoned. My only suggestion is to include some snapshots in the SI of the non-canonical binding events. Currently, only a graph of distance is shown in Figure S1.

Minor points

Page 3: "was identified determined by" - remove one identified/determined

Page 6: "head one bind" - binds

Figure 4C: the time axis is incorrect; it's stated to be ns but is actually microseconds.

**Have all data underlying the figures and results presented in the manuscript been provided?**

Reviewer #1: Yes

Reviewer #2: **No: **It sounds like the authors are planning share simulation trajectories on Zenodo, which is great.

Reviewer #3: Yes

Reviewer #4: None

PLOS authors have the option to publish the peer review history of their article (what does this mean?). If published, this will include your full peer review and any attached files.

Reviewer #1: No

Reviewer #2: No

Reviewer #3: **Yes: **Arieh Warshel

Reviewer #4: No
---

## [Decision Letter · Decision Letter 1]

28 Jul 2021

Dear Prof. Sansom,

Thank you very much for submitting your manuscript "Membrane-binding mechanism of the EEA1 FYVE domain revealed by multi-scale molecular dynamics simulations" for consideration at PLOS Computational Biology. As with all papers reviewed by the journal, your manuscript was reviewed by members of the editorial board and by several independent reviewers. The reviewers appreciated the attention to an important topic. Based on the reviews, we are likely to accept this manuscript for publication, providing that you modify the manuscript according to the review recommendations.

The revised version addresses virtually all comments by the reviewers.

However, as pointed out by reviewer #2 there are several public and free repositories that allow depositions up to 50GB (and even more if one has a good reason), and since it is an explicit PLoS editorial policy that snapshots from trajectories must be available in a public FAIR repository ( as described e.g. in https://journals.plos.org/ploscompbiol/article?id=10.1371/journal.pcbi.1006649 ), this must be added before we can finally accept the paper. The exact repository used does not matter - a few possible examples include Zenodo, OSF, Dryad or NOMAD, so the authors can choose whatever suits them best.

Sincerely,

Erik Lindahl

Guest Editor

PLOS Computational Biology

Nir Ben-Tal

Deputy Editor

PLOS Computational Biology

[LINK]

Thank you for the revised version, which addresses virtually all comments by the reviewers.

However, as pointed out by reviewer #2 there are several public and free repositories that allow depositions up to 50GB (and even more if one has a good reason), so to adhere to the PLoS data sharing policies (as described e.g. in https://journals.plos.org/ploscompbiol/article?id=10.1371/journal.pcbi.1006649 ) representative downsampled trajectories must be deposited with a link in the data availability section before we can finally accept the paper.

The exact repository used does not matter - a few possible examples include Zenodo, OSF, Dryad or NOMAD.

Reviewer's Responses to Questions

**Comments to the Authors:**

Reviewer #2: The authors have addressed my main comments, and I think the paper can be accepted for publication.

Regarding data availability, the authors correctly point out that there is no standard repository for MD simulations. However, data of any type can be deposited on Zenodo, which I believe has a limit of 50 GB per submission, so I don't see why simulation trajectories could not be uploaded there (or to a similar repository). If the size limit is an issue, the trajectories could be downsampled or otherwise reduced.

A minor additional comment:

Figure 3 caption reads "from the simulation"; since there are now 3 atomistic simulations, shouldn't this read "a representative simulation" or similar?

Reviewer #3: The curent version followed my comment

Reviewer #4: Looks great!

**Have the authors made all data and (if applicable) computational code underlying the findings in their manuscript fully available?**

Reviewer #2: **No: **See comment to authors.

Reviewer #3: Yes

Reviewer #4: None

PLOS authors have the option to publish the peer review history of their article (what does this mean?). If published, this will include your full peer review and any attached files.

Reviewer #2: No

Reviewer #3: No

Reviewer #4: No

Figure Files:

Data Requirements:

Reproducibility:

References:

---

## [Editor Report · Decision Letter 2]

14 Sep 2021

Dear Prof. Sansom,

We are pleased to inform you that your manuscript 'Membrane-binding mechanism of the EEA1 FYVE domain revealed by multi-scale molecular dynamics simulations' has been provisionally accepted for publication in PLOS Computational Biology.

Best regards,

Erik Lindahl

Guest Editor

PLOS Computational Biology

Nir Ben-Tal

Deputy Editor

PLOS Computational Biology

---

## [Editor Report · Acceptance letter]

17 Sep 2021

PCOMPBIOL-D-21-00195R2 

Membrane-binding mechanism of the EEA1 FYVE domain revealed by multi-scale molecular dynamics simulations

Dear Dr Sansom,

I am pleased to inform you that your manuscript has been formally accepted for publication in PLOS Computational Biology. Your manuscript is now with our production department and you will be notified of the publication date in due course.

With kind regards,

Andrea Szabo
